# Machine learning assisted interferometric structured illumination microscopy for dynamic biological imaging

Edward N. Ward [1], Lisa Hecker[1], Charles N. Christensen[1], Jacob R. Lamb[1], Meng Lu[1], Luca Mascheroni[1], Chyi Wei Chung [1], Anna Wang [2], Christopher J. Rowlands [3], Gabriele S. Kaminski Schierle [1] & Clemens F. Kaminski [1] ✉

Structured Illumination Microscopy, SIM, is one of the most powerful optical imaging methods available to visualize biological environments at subcellular resolution. Its limitations stem from a difficulty of imaging in multiple color channels at once, which reduces imaging speed. Furthermore, there is substantial experimental complexity in setting up SIM systems, preventing a widespread adoption. Here, we present Machine-learning Assisted, Interferometric Structured Illumination Microscopy, MAI-SIM, as an easy-to-implement method for live cell super-resolution imaging at high speed and in multiple colors. The instrument is based on an interferometer design in which illumination patterns are generated, rotated, and stepped in phase through movement of a single galvanometric mirror element. The design is robust, flexible, and works for all wavelengths. We complement the unique properties of the microscope with an open source machine-learning toolbox that permits real-time reconstructions to be performed, providing instant visualization of super-resolved images from live biological samples.

Super-resolution microscopy has enabled the observation of increasingly smaller features in biological specimens, but many challenging problems remain, for example in the imaging of intracellular transport processes and small organelle interactions in live cells. Reasons for this are limitations in acquisition speed, restrictions on fluorescent labels or required buffers, and the need for the use of illumination intensities compatible with viewing live samples. Structured Illumination Microscopy (SIM) stands out as a technique for sub-diffraction imaging of live biological specimens as it strikes an optimal balance in resolution and imaging speed at light intensities where phototoxicity levels are tolerable[1,2]. As SIM uses widefield detection and requires only a linear fluorescent response, large fields of view (FOVs) can be imaged at low excitation powers and high speeds.

In SIM, a spatially modulated illumination pattern is projected onto the sample, which mixes with high spatial frequency content in the sample structure to generate beat frequencies. The low-frequency beat patterns, or Moiré fringes, contain high-resolution sample information, which can be reconstructed into super-resolved images. In most implementations of SIM, a striped excitation pattern with sinusoidal modulation is used, containing a single spatial frequency. The resulting high-resolution image is reconstructed from a series of nine sequentially recorded images corresponding to three linear translations of the pattern (phase stepping) and three rotation steps of 120° to obtain isotropic resolution enhancement. In such a scheme, spatial frequencies in the sample are downshifted by an amount equal to the spatial frequency of the striped illumination. As the excitation pattern is itself limited by the frequency passband of the imaging system, $2NA/\lambda$, spatial frequencies of twice this limit can be downshifted into the passband, leading to an approximately twofold increase in resolution.

[1]Department of Chemical Engineering and Biotechnology, University of Cambridge, Cambridge, UK. [2]Department of Physics, Oxford University, Oxford, UK. [3]Department of Bioengineering, Imperial College London, London, UK. ✉e-mail: cfk23@cam.ac.uk

A number of SIM variants have been developed which differ in the way patterns are generated and projected onto the sample. In the earliest implementations of SIM, illumination patterns were created with fixed diffraction gratings[2]. The excitation was diffracted into multiple beams which were recombined in the sample plane to form striped interference patterns. However, the fixed periodicity of the grating corresponds to a fixed spatial frequency. In practice, this means that the method could only be optimized for a single wavelength. For multicolor imaging this requires the use of multiple gratings, which is practically complex, or it must be accepted that the method will perform sub-optimally for other wavelengths. Pattern changing was achieved by mechanical rotation and translation of the diffraction grating, limiting operation at high speed. Later implementations make use of Spatial Light Modulators (SLMs) based on either liquid crystal technologies[3–5] or Digital Micro-mirror Devices (DMDs)[6–8] for pattern generation, which increases acquisition speed by an order of magnitude. However, these methods have their own limitations. For example, as is the case for diffraction gratings, new patterns have to be generated for every color to obtain the best resolution, and again channels must be recorded sequentially. Additionally, the discrete number and finite size of pixels restrict the possible periodicities of the pattern and introduce unwanted diffraction orders, which reduce photon efficiency[3]. Typically light throughput is limited to a few percent of the input energy, requiring higher power lasers for illumination. The Michelson implementation also leads modulation patterns over large fields of view, not restricted by the physical size of the SLM/DMD. Finally, all of these methods are highly sensitive to the polarization state of the incoming light, and this must be controlled in synchronicity with the pattern orientation to maximize pattern contrast in the sample[9].

Here, we present an interferometric method to generate illumination patterns for SIM: Machine learning Assisted Interferometric SIM, MAI-SIM. The method makes use of an interferometer to generate fringes which are projected into the sample. The instrument projects patterns of optimal spatial frequency into the sample, irrespective of excitation wavelength. The method is fast, low in cost, and straightforward to implement. We demonstrate simultaneous SIM imaging in three colors over large FOVs (44 μm × 44 μm) at several frames per second recording speed. We maximize the potential of MAI-SIM with a suite of freely available software tools for image reconstruction. The method is constructed using open design principles. All hardware and software modules to implement MAI-SIM are fully described, for anyone to implement on standard fluorescence microscopy frames. Finally, we demonstrate the power of MAI-SIM through application in live cell samples, where we image dynamic organelle-organelle interactions in three colors. MAI-SIM records, renders, and visualizes super-resolution output in real time.

## Results

### Hardware implementation

The core component of the presented microscope is a Michelson interferometer, where wavelength-dependent wedge fringes are projected onto the sample. A single high-speed mirror galvanometer is used to address one of three pairs of complementary 0.5" mirrors (Fig. 1A). Each pair of mirrors forms fringes of constant thickness whose period is governed by the wavelength of the excitation light and the inclination of the mirrors with respect to one another. An image of these fringes, corresponding to an image of the surface of these mirrors, is subsequently relayed onto the sample (Supplementary Figs. 1, 2).

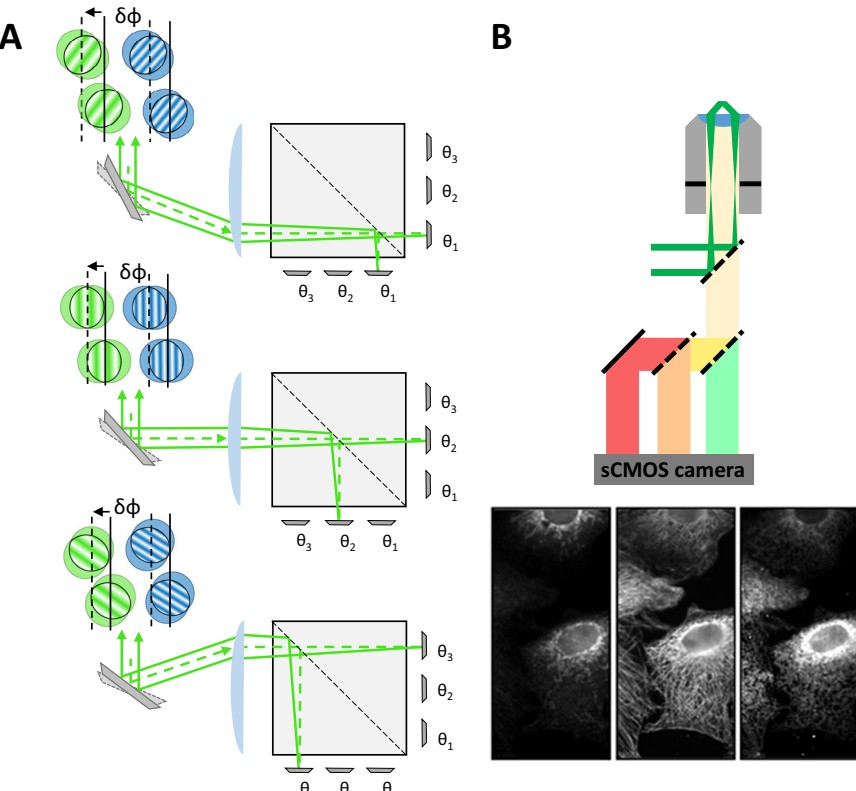

**Fig. 1 | Working principle of MAI-SIM. A** Schematic of the interferometric design. Three small mirror pairs, each forming a Michelson interferometer, are tilted to form interference patterns at angles, $\theta_{1-3}$, separated by $2\pi/3$. Large steps of the scanning mirror change the mirror pair addressed by the incoming beam (dashed line) and hence the rotation of the fringe pattern. Small steps of the scanning mirror laterally displace the position of the interference fringes and cause a phase shift,

$\delta\phi$, in the excitation pattern. Different wavelengths (blue and green patterns) produce fringes with different spacing and hence an unequal phase shift.
**B** Schematic of detection optics. Using several dichroic mirrors and emission filters, the three-way image splitter allows for the simultaneous display of up to three color channels on the camera chip. For full details on the optical setup and a rendering of the layout, see Supplementary Fig. 1.

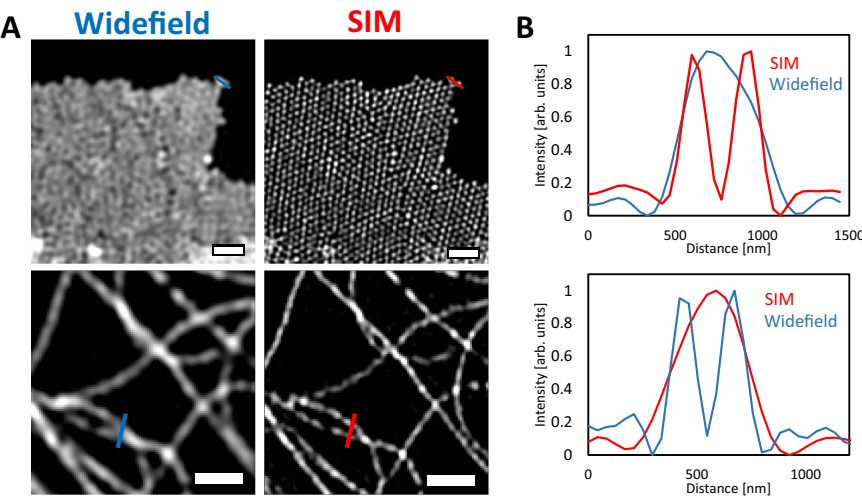

**Fig. 2 | Demonstration of the MAI-SIM on samples excited at a wavelength of 561 nm.** In the deconvolved widefield images, adjacent 200 nm beads ($N = 4$, top) and entangled microtubule filaments ($N = 3$, bottom) remain unresolved but are clearly distinguishable in the super-resolved image. Images were reconstructed using the inverse matrix method[14]. Scale bars are 1 μm. Source data are provided as a Source Data file.

Furthermore, by adjusting the tilt of each mirror individually, fringes are produced such that moving from one mirror pair to the next rotates the fringe pattern around the optical axis. Thus, stepping from one mirror pair to another (Fig. 1) rotates the pattern by 120°. Finally, phase shifting of the pattern is achieved using small galvanometer steps to translate the beam across each mirror pair, which yields a fixed lateral shift in the projected image plane. This implementation greatly reduces the complexity associated with interferometric systems—such as that employed in the latest iteration of the OMX SIM microscope—which require multiple moving components and piezo elements for phase stepping[10–12]. Additionally, as the interfering beams share a nearly common path, the phase stability of the system is greatly increased over implementations where the arms of the interferometer are more spatially separated. Because the fringes are coherently formed, it is critical that the coherence lengths of the lasers used in the setup exceed the maximum path difference between the interfering beams. Good temporal coherence is thus important, in contrast to SLM-based SIM systems. In our setup, mirror pairs are on individual translation stages so that the path length difference can be precisely controlled to obtain optimal fringe contrast.

A key issue with SIM in high Numerical Aperture (NA) imaging systems is that the modulation depth of the pattern, which governs the strength of the super-resolution information collected, depends on the polarization orientation of the incident light. For high pattern contrast, s-polarization must be maintained for all pattern rotations. In many SIM systems, this is achieved through the use of electro-optic liquid crystal devices which must be synchronized to the pattern generation process and the wavelength chosen. Here we chose to use a wedged polarizer placed in the Fourier plane of the microscope as pioneered by Heintzmann et al.[13] To maintain the correct linear polarization at the back aperture of the objective, a matched pair of quad-band dichroic mirrors was used to compensate for the inherent ellipticity introduced by such mirrors. The detection path of the system was equipped with a commercial three-way image splitter, which included dichroic mirrors and emission filters to enable the simultaneous detection of three different fluorophores on a single high-speed camera (Fig. 1B). The synchronization of the individual components is simpler than for many other SIM systems, as fewer active elements need to be considered.

## Instrument characterization

To quantify the lateral resolution improvement, 100 nm fluorescent spheres were imaged at an excitation wavelength of 488 nm. We determined the full-width-half-maximum (FWHM) of the intensity profiles for individual point emitters by fitting the images to a 2D Gaussian profile (Supplementary Fig. 6). The mean FWHM of 94 emitters was measured as 200 nm for deconvolved widefield images and 120 nm for reconstructed MAI-SIM images, which is as expected for the spatial frequency of the fringe pattern. The resolution improvement can also be seen using the Fourier Ring Correlation (FRC) method where a 1.77 fold increase is resolution is apparent compared to the unprocessed widefield images (Supplementary Fig. 7). The performance of the system was tested using fluorescent microspheres with 200 nm diameter as well as mCherry-labeled tubulin, both of which have well-defined structures. The samples were imaged with an excitation wavelength of 561 nm, and reconstructed using an inverse matrix approach implemented in MATLAB[14]. Fig. 2 compares the reconstructed MAI-SIM data with the corresponding Wiener-filtered widefield images. While the latter provides suppression of out-of-focus light and an enhanced visual appearance, MAI-SIM is capable of significantly enhancing the resolution, which is crucial to distinguish individual particles and filaments.

## The use of neural networks enhances image reconstruction in MAI-SIM

In MAI-SIM, patterns of multiple colors can be projected into a sample plane simultaneously. Phase stepping is achieved through small lateral shifts across specific mirror pairs (see Fig. 1A). As the fringe spacing is different for each wavelength, this means that not all colors will experience the same phase shift for the same lateral displacement. In practice, this means that the transverse pattern shifts required to reconstruct images may not correspond to the ideal $2\pi/3$ phase steps. In sequential imaging this is not a problem, since the shifts can then be adjusted to be optimal for each individual wavelength. For the case of uneven phase steps, traditional Fourier-based reconstruction techniques do not perform optimally[10,14–17]. We have compared the performance of several widely used reconstruction methods, specifically, the FairSIM-imageJ plugin[17], an inverse matrix method[14], an iterative Blind SIM method[18], and finally our Machine Learning-SIM (ML-SIM) approach[19]. FairSIM is an easy-to-implement plugin with well-documented and versatile pre- / post-processing techniques[20] which are familiar to existing SIM users. In principle, FairSIM offers the capability of estimating phase step sizes from SIM patterns; however, this capability only works for even phase steps. The inverse matrix method implements a more sophisticated method for phase estimation which is capable of performing reconstructions with uneven phase shifts. Its disadvantage compared to FairSIM is its limited versatility in pre- and

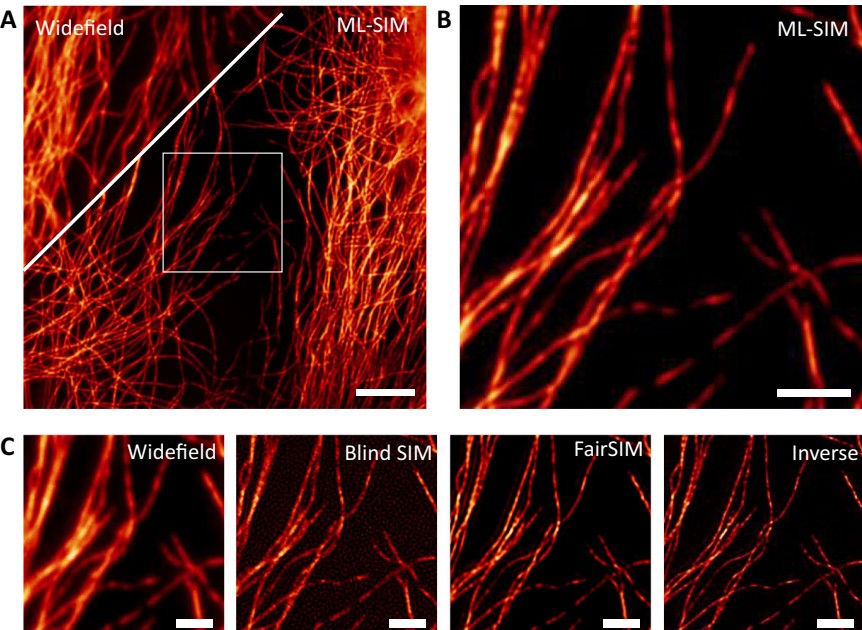

**Fig. 3 | Reconstructions of MAI-SIM data from fixed cells.** Fixed COS7 cells containing beta-tubulin labeled with secondary antibodies conjugated to Alexa568 dye. **A** Comparison of diffraction limited widefield images and ML-SIM reconstruction over the full field of view. Scale bar is 5 μm. **B** Machine Learning (ML-SIM) reconstruction for magnified region indicated in **A**. Scale bar is 1 μm. **C** From left to right: widefield, Blind SIM, FairSIM, and inverse matrix reconstructions over the same region. In all reconstruction methods, a resolution increase is apparent compared to the widefield image. In the case of ideal phase shifting, the FairSIM and inverse matrix reconstructions have performed comparably. The resolution increase achieved by the iterative reconstruction method is less than that of the other techniques and there is significant amplification of background noise. Scale bars are 1 μm, $N = 4$.

post-processing modalities. Iterative Blind SIM reconstruction is a fundamentally different approach and was developed for the reconstruction of SIM data acquired under random excitation patterns[21–23]. This makes the technique inherently agnostic to the specifics of phase steps used so long as a sufficiently good initial estimate for the sample image can be made[18]. For our ML-SIM, even steps are also not necessary and the method is specifically optimized for the reconstruction of interferometric SIM data, although it works equally well with all other implementations for SIM.

ML-SIM is a deep neural network consisting of approximately 100 convolutional layers, similar in architecture to residual channel attention networks[24]. These are designed for single-image upsampling, where here the architecture is modified to take nine raw SIM frames as input from which the image is reconstructed. For this study, our previously published approach was developed to address the specific challenges of MAI-SIM. The model is trained on synthetically generated SIM data with a full physical model of the SIM process, which includes experimental factors such as spherical aberrations, Poisson noise, and finite pattern contrast (Supplementary Fig. 8). Data are simulated for various point spread functions, illumination angles, phase steps, and noise sources. The imaging parameters are randomized, such that the model learns to be robust irrespective of the specifics of a particular SIM imaging setup. The quality of these reconstructions is compared in Figs. 3 and 4. Figure 3 demonstrates the performance on interferometric SIM data from a sequential acquisition of multiple color channels. This permits the phase shifts to be optimized individually for all wavelengths. As expected under these conditions, the performance of the inverse matrix and FairSIM is comparable, because the assumption of equidistant phase steps of 2π/3 is valid. In contrast, while a resolution improvement over the widefield image is still apparent, the Blind SIM approach introduces noticeable reconstruction artifacts, associated with the amplification of noise. Figure 4, on the other hand, compares the performance of these reconstruction techniques on data acquired during the simultaneous acquisition of all color channels. Here, the phases were optimized for 561 nm excitation, i.e., with equidistant phase steps separated at 2π/3;

however, the images shown are for the 488 nm excitation channel, where phase steps are no longer equidistant and differ significantly from 2π/3. As seen, ML-SIM outperforms the classical reconstruction methods by a considerable margin. The algorithm handles uneven phase steps and deals better with data acquired at low signal-to-noise ratios (SNRs). This improved performance on data with uneven phase steps can also be seen in the reconstruction of simulated data with precisely controlled phase steps where the. (Supplementary Fig. 9). For the imaging of dynamically changing and light-sensitive samples, where short camera exposure times and low excitation intensities have to be used, the ability to handle noise and uneven phase stepping presents very significant advantages. We quantified the relative performance of the different methods on experimental data using NanoJ analysis software[25] (Supplementary Fig. 10, and Supplementary Table 1).

In addition to an increased robustness to noise and pattern phase errors, the use of machine learning for SIM enables image reconstruction that is considerably faster than Fourier or iterative methods. We leverage this advantage to demonstrate on-the-fly reconstructions of live generated SIM data. On an entry-level graphics card, image acquisition, reconstruction, and visualization of up to three color channels can be performed simultaneously at a frame rate of 1 Hz over a 44 μm × 44 μm FOV (Supplementary Fig. 11). A video demonstration of live MAI-SIM imaging showing the responsiveness and ease of use of the method, even for non-experts is available in the figshare repository for the work [https://figshare.com/projects/MAI-SIM/140008].

## MAI-SIM enables imaging in multiple color channels

To assess the performance of MAI-SIM for imaging in multiple colors, we first imaged labeled fluorescent microspheres and intracellular organelles in fixed COS-7 cells. Figure 5 shows multicolor microspheres excited at 488 nm, 561 nm, and 647 nm, respectively. Data in the three channels were acquired sequentially so that phase steps could be optimized for every color. In this case, classical reconstruction algorithms work well. Images shown were reconstructed using the inverse matrix method and are clearly better resolved than the

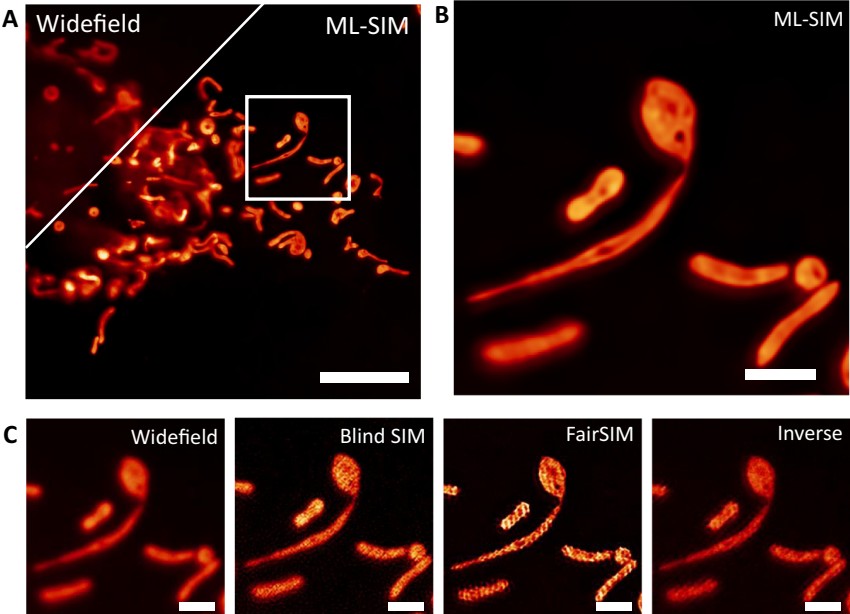

**Fig. 4 | Machine learning outperforms classical SIM reconstruction algorithms for the analysis of data acquired with uneven phase steps.** Reconstructions of data from GFP-labeled mitochondria in live samples with data acquired during simultaneous three color imaging (only one color channel shown here). GFP was excited at 488 nm, but phase steps were optimized for the separate 561 nm color channel, leading to phase steps that differ from 2π/3. **A** Comparison of diffraction limited imaging and Machine Learning (ML-SIM) over the full field of view. Both resolution and background rejection are enhanced in the ML-SIM reconstruction. Scale bar is 5 μm. **B** ML-SIM reconstruction of magnified view indicated in **A**. The ML-SIM method is unaffected by the uneven phase stepping and resolution is improved without reconstruction artifacts. Scale bar is 1 μm. **C** Left to right: widefield, Blind SIM, FairSIM, and inverse matrix reconstructions for indicated region. The uneven phase stepping leads to significant reconstruction artifacts, including striping artifacts and honeycombing. While the inverse matrix approach remains free from striping artifacts, additional honeycombing is apparent due to low Signal-to-Noise ratios. The blind SIM reconstruction shows a limited increase in resolution over the widefield image and amplification of background noise. Striping artifacts are also present as a result of a poor initial estimate of the sample. Scale bars are 1 μm, *N* = 3.

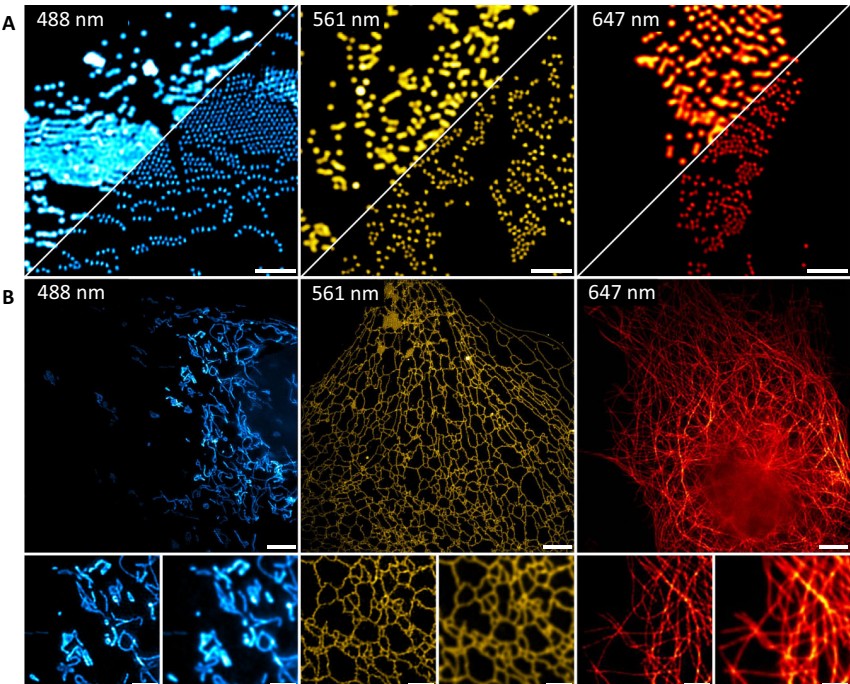

**Fig. 5 | Multicolor imaging capability of the developed instrument. A** 200 nm TetraSpeck microspheres imaged with excitation/emission wavelengths of 488/515 nm, 561/580 nm, and 647/680 nm, respectively (*N* = 1). In each panel, the top left corner shows the deconvolved widefield image, and the lower right corner shows the inverse matrix SIM reconstructions. Scale bars are 2 μm. **B** Multicolor imaging of cellular organelles with different staining strategies. 488 nm excitation: fixed mitochondria (GFP, *N* = 1); 561 nm excitation: fixed endoplasmic reticulum (immunostained Alexa568, *N* = 1); 647 nm excitation fixed beta-tubulin (immunostained Alexa647, *N* = 1). Scale bars are 5 μm (top/middle) and 500 nm (bottom).

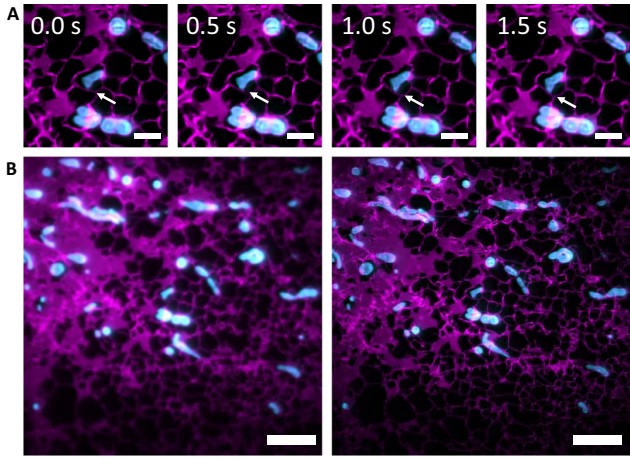

**Fig. 6 | Simultaneous imaging of live cells in multiple color channels.** COS7 cells were stained with mitotrackerGreen and transfected with mApple-Sec61b-C1 to label the endoplasmic reticulum (ER). **A** Timelapse images of mitochondria (blue) embedded in rapidly reorganizing ER networks (magenta). The arrow indicates a contact site between the mitochondria and the ER. Contact is seen to be maintained as the structures reorganize. Images were taken at 0.5 s intervals. Scale bar is 2 μm. **B** Comparison of diffraction limited (left) and SIM (right) images. Simultaneous imaging of multiple channels permits interactions between rapidly moving small structures to be investigated without temporal (and thus spatial) offsets. Scale bar is 5 μm, $N = 3$.

deconvolved widefield images shown for comparison. Figure 5B shows fixed GFP expressing mitochondria (blue), fixed ER labeled with Alexa568 immunostaining (yellow), and fixed microtubules labeled with Alexa647 immunostaining (red). In all three images, filament networks, bundles, and junctions can be resolved at sub-diffraction resolution. This demonstrates that artifact-free, high-resolution SIM can be obtained for a variety of samples and labeling methods with classical reconstruction methods.

Finally, we assess the use of MAI-SIM for the investigation of live biological samples. The ability to image multiple color channels simultaneously without temporal offset is crucial for the investigation of dynamic interactions between different biological structures, yet is not available with existing SIM modalities. To demonstrate this capability, live COS7 cells were labeled with mitotrackerGreen, mApple-Sec61b-C1 conjugate, and SiR-lysosome. Figure 6 shows a visualization of mitochondria-ER interactions during ER reorganization. The simultaneous acquisition of SIM frames in the different color channels permits the observation of ER structures wrapping dynamically around the mitochondrial periphery without any temporal offsets. The arrow indicates the location of a contact site between the ER (magenta) and a mitochondrion (cyan). As the mitochondrion moves, contact between the two structures is seen to be maintained, an observation that could not have been made with sequentially performed SIM imaging (see video 2 in SI).

It has recently been suggested that lysosomes play an active role in the rapid reorganization of ER network topology, providing the pulling force for rapid tubule formation and the forming of ER network connections[26]. Fig. 7 shows the unique capability of MAI-SIM to track such dynamic cellular processes. In the images shown, lysosomes appear in yellow, and the ER network in magenta. Using MAI-SIM with the detection splitting optics (Fig. 7A) leads to perfect co-registration of the two color channels in time, and an effective doubling of the achievable frame rate with respect to sequential SIM imaging. The bottom row shows the same event, but this time using the camera frames in the data that mimic a sequentially recorded SIM image (Fig. 7B). The camera frame rate was 20 frames per second over a 44 μm × 44 μm FOV; the panels shown in the sequences are cropped

regions from the full frame view (Fig. 7C). Two events are seen taking place. As demonstrated in Fig. 7, a lysosome (indicated with the blue marker) is seen to pull an ER tubule towards another region of the ER but ultimately fails to make a connection, the lysosome detaches, and the tubule subsequently retracts. A fast-moving lysosome (green marker) is seen to travel downwards, cleaving an existing tubule (white arrow) to form two new tubules. Both of these events would have been hard to visualize or interpret on sequentially recorded SIM data. We simulated sequentially recorded SIM by analyzing the nine frames required to generate a SIM image for the lysosome channel and the following nine frames for the ER channel. The sequencing of the frames thus simulates a traditional SIM experiment and leads to a halving of the effective frame rate. The corresponding data are shown in Fig. 7B. Neither of the two phenomena described above could have been correctly interpreted in the sequential data, because the correlation between the organelle locations is lost. MAI-SIM is thus an ideal technology to study organelle biology such as these transient interactions between lysosomes and the ER[26].

## Discussion

We have developed a fast, structured illumination microscope which uses two beam interferometry to generate illumination patterns. The method is ideally suited for the investigation of fast biological processes in multiple color channels. The pattern generation process is optimal for any color, and patterns can be projected into the sample simultaneously. Combined with a multi-channel image detection system, this allows for sub-diffraction resolution imaging in multiple colors simultaneously while maintaining optimal resolution for all excitation wavelengths. We demonstrate such simultaneous imaging on fast-moving cellular structures at a speed of 3 Hz, close to the maximum frame rate of the camera for full FOV. The maximum imaging speed of our system is limited by the settling time galvanometric mirror element used. Because of its size and associated inertia it requires several tens of milliseconds to settle between mirror-to-mirror jumps. The system could be readily adapted to make use of a smaller, and faster, mirror element so long as a de-magnified image of the objective back focal plane does not exceed the size of the mirror. With recently available MEMS mirrors, settling should not be the limiting parameter of imaging speed and frame rates only limited by camera speed would then be available for MAI-SIM.

The process is straightforward and cost-effective to implement on existing widefield microscope frames. Furthermore, our method does not fundamentally differ in the way it shifts high frequency sample information into the pass band of a microscopy from conventional SLM or DMD based SIM methods. Hence, all classical reconstruction methods are, in principle, applicable to our method if phase steps in the data generation process are sufficiently controlled. However, there are unique aspects to interferometric SIM which make the use of traditional reconstruction algorithms challenging when imaging in multiple colors simultaneously. We address this problem by modeling the physical process of interferometric SIM to generate in silico data that can be used to train a neural network. The hardware and software together provide a powerful and easy-to-use high-speed SIM platform.

Our approach allows for the reliable reconstruction of low-SNR SIM images in which pattern phase shifts cannot be precisely controlled, for example during simultaneous data acquisition in multiple color channels where the system has been optimized for the central excitation wavelength. The analysis package is also applicable for use in traditional SIM modalities but has the added advantage of speed. Once the network is trained, reconstruction and visualization can be achieved in multiple colors with reconstruction frame rates exceeding 1 Hz (higher for smaller FOVs). MAI-SIM data is acquired with no temporal offsets between the channels, enabling the observation of

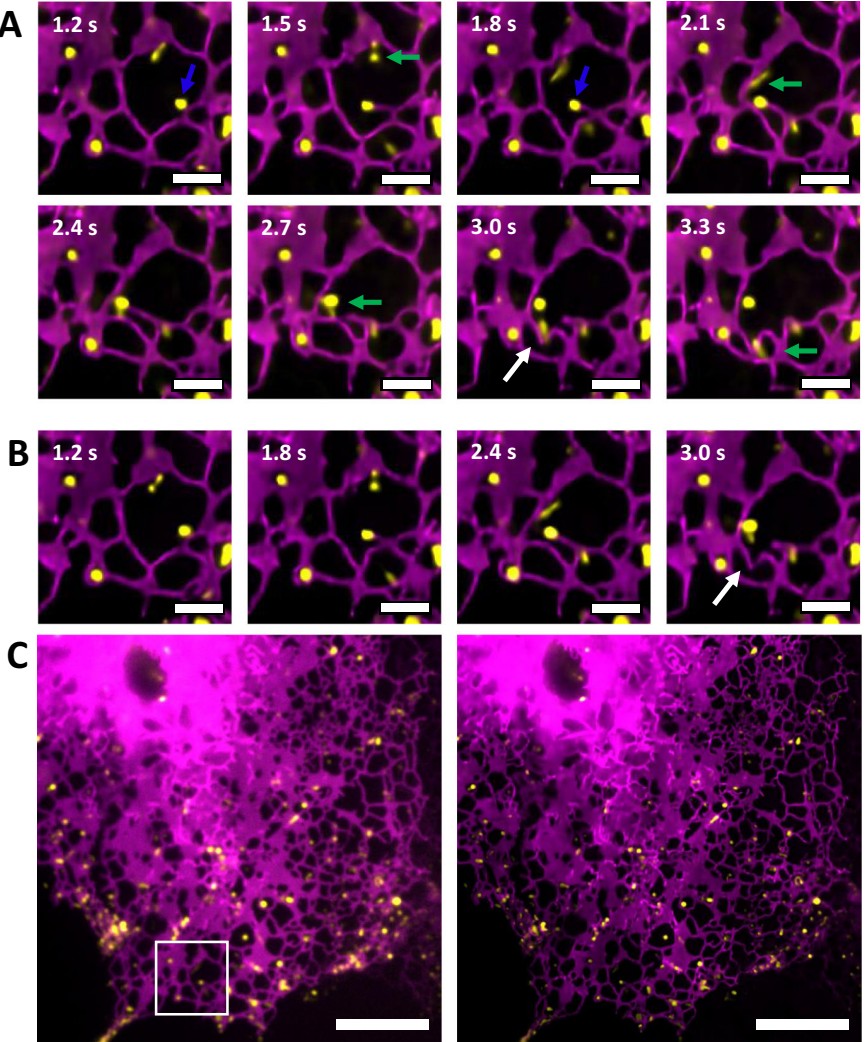

**Fig. 7 | Simultaneous imaging with MAI-SIM improves temporal resolution and image reliability.** COS7 cells stained with SiR-lysosome (yellow) and transfected with mApple-Sec61b-C1 to label the endoplasmic reticulum (ER) (magenta). **A** Time series of two ER network reorganization events visualized with MAI-SIM. One lysosome (blue marker) can be seen to pull a new filament from the network which fails to make a connection and is subsequently released. A second lysosome (green marker) moves rapidly along the ER network and cleaves an ER filament (white arrow) to form two new filaments. Scale bar is 2 µm. **B** The same ER network events visualized in a sequential acquisition. In this case, the temporal offset between the color channels means that the fast cleaving event (white arrow) appears independent of lysosome transport. Scale bar is 2 µm. **C** Full field of view of ER network in widefield (left) and MAI-SIM (right) imaging. The region showing the time series data is indicated by the white square. Scale bar is 10 µm, $N = 5$.

organelle-organelle interactions in real time. Live analysis and visualization of MAI-SIM data is possible. We demonstrate the method in the study of interactions between mitochondria, ER, and lysosomes, and highlight how their interactions control and affect ER remodeling processes. The presented system is compact and, with only one actively moving component, does not require complex synchronization procedures, electronic control of the polarization state, or use of expensive and photon-inefficient spatial light modulators. MAI-SIM could easily be extended to permit TIRF-SIM imaging[27,28] and an extension of the method to enable 3D SIM would be possible through the addition of a third arm to the interferometer (see SI Fig. S12 for a proposal on how this might be achieved)[29–31]. The microscope is suitable for use in a variety of biological applications and is a powerful tool for multicolor imaging of living cells and intracellular organelle interactions in real time. We therefore envisage MAI-SIM to be an attractive alternative to commercial and SLM-based SIM instruments, making high-resolution imaging available for a wide scientific community.

## Methods

### Hardware setup

The setup was based around a commercial microscope frame (Olympus IX73) equipped with a 1.2 NA water immersion objective lens. Three laser lines (647 nm, 561 nm and 488 nm) were combined co-axially using dichroic mirrors. The beams were expanded by 10 times and entered the interferometer from a single galvanometric mirror. The interferometer was based around a single two-inch 50:50 beamsplitter cube and three pairs of half-inch mirrors. The returning beamlets were descanned by the galvanometric mirror and relayed into the microscope through two 4-$f$ lens assemblies. A wedged polarizer was placed in a plane conjugate to the back focal plane of the objective lens (Supplementary Fig. 1A). A pair of quad-band dichroic mirrors were used to separate the fluorescence signal from the excitation laser light while maintaining ideal illumination polarization. The fluorescence emission was further separated into three wavelength ranges 503–551 nm; 573–623 nm; and 658–800 nm which were imaged onto different regions of the camera sensor (PCO, Edge 4.2 bi).

## Image acquisition

Image data were acquired through the open-source software micromanager (version 2.0.1). Hardware control and camera triggering were performed through a custom-made interface built using LabVIEW (version 21.0.1). Briefly, the camera was triggered in a level trigger mode, with exposure time controlled externally using the user interface. In this way, camera trigger was synchronized with laser control and mirror switching. Details of the precise timings used can be found in the code repository for the work. [https://github.com/LAG-MNG-CambridgeUniversity/MAI-SIM][32].

## Super-resolution image reconstruction

Images were processed using custom code and the open-source software, ImageJ (version 2.35). Classical SIM reconstructions were performed in FairSIM, which was implemented within the ImageJ environment. Background subtraction was used to remove noise before processing and simulated optical transfer functions were used for filtering. A Weiner filter was used both before and after reconstruction with a Weiner factor chosen empirically for each image. Zero-order removal and stripe suppression were applied after reconstruction. Global background subtraction was applied after reconstruction.

Inverse matrix reconstructions were performed in MATLAB 2021b using a script developed from the original implementation[14]. Weiner filtering was used on the input and output with the factor chosen empirically for each image. No background subtraction, zero-order removal or stripe suppression were used for the data presented.

Machine learning reconstructions were performed using a custom-made program for Python (version 3.9). The machine-learning method is parameter-free and no pre- or post-filtering were used. The code used to reconstruct the images and pre-trained neural networks are available from the GitHub repository [https://github.com/LAG-MNG-CambridgeUniversity/MAI-SIM][32].

## Bead preparation

Bead samples were prepared according to manufacturer protocols. Briefly, 200 nm and 100 nm bead monolayers (ThermoFisher, TetraSpeck) were formed by air-drying onto a 12 mm coverglass 5 μl of bead suspensions at $2.3 \times 10^7$ and $1.8 \times 10^8$ particles/mL, respectively. The samples were then mounted in 15 μl DI water and secured to glass slides with spacers before being sealed with nail polish.

## Fixed cell imaging

COS-7 cells (from monkey kidney tissue) were plated onto 13 mm round coverslips in a 4 well plate at 20,000 cells per well and cultured under standard conditions (37 °C, 5% $CO_2$) in minimum essential medium (Sigma Aldrich) supplemented with 10% fetal bovine serum (Gibco) and 2 mM L-lutamine (GlutaMAX, Gibco). The next day, cells were transfected using CellLight Mitochondria-GFP (Invitrogen), dispersing 8 μL of the transfecting reagent in the cell culture medium and incubating the cells for 24 h under standard conditions. Cells were fixed by incubation with 4% methanol-free formaldehyde and 0.1% glutaraldehyde in cacodylate buffer (pH 7.4) for 15 mins at room temperature, washed three times with PBS, and then permeabilized by incubation with a 0.2% solution of saponin in PBS for 15 mins. Unspecific binding was blocked by incubating with 10% goat serum and 100 mM glycine in PBS and 0.2% saponin for 30 mins at room temperature. Without washing, the samples were incubated with the primary antibodies mouse anti-beta-tubulin (Abcam, ab131205), rabbit anti-calnexin (Abcam, ab22595) diluted 1:200 in PBS containing 2% BSA (bovine serum albumin) and 0.2% saponin overnight at 4 °C. After three washes in PBS, the samples were incubated with the secondary antibodies goat anti-mouse conjugated to AlexaFluor647 (Invitrogen, A21235), goat anti-mouse conjugated to AlexaFluor568 (Invitrogen, A21124) or goat anti-rabbit conjugated to AlexaFluor568 (invitrogen, A11036 or) diluted 1:400 in PBS containing 2% BSA and 0.2% saponin

for 1 h at room temperature in the dark. Samples were then washed 3 times with PBS. Finally, the coverslips were mounted on glass slides using a Mowiol-based mounting medium.

## Live cell imaging of ER-lysosome-mitochondria dynamics

COS-7 (American Type Culture Collection, ATCC CRL-1651) cells were plated into 8 well plates (Ibidi) at 5,000 cells per well and cultured under standard conditions (37 °C, 5% CO2) in Dulbecco's Modified Eagle's Media (DMEM, ThermoFisher) supplemented with 10% fetal bovine serum (Gibco) and 2 mM L-glutamine (GlutaMAX, Gibco). 24 h before imaging, cells were transfected with the plasmid construct mApple-Sec61β-C1 (Addgene plasmid #90993) using Lipofectamine 2000 (ThermoFisher) according to the manufacturer's protocol. 20 h prior to imaging cells were stained with SiR-Lysosome (Cytoskeleton Inc.) and Verapamil (Cytoskeleton Inc.) at 1 μM and 10 μM respectively. 30 mins prior to imaging cells were stained with MitoTracker Green FM (ThermoFisher) at 250 nm and washed immediately before imaging with DMEM. Imaging was performed with a stage-top incubator (OKOLab) at 37 °C and 5% CO2.

## Statistics and reproducibility

Multiple repeats were acquired from single samples. The number of repeats used, $N$, is indicated in figure descriptions. Similar results were observed for all repeats.

## Reporting summary

Further information on research design is available in the Nature Portfolio Reporting Summary linked to this article.

## Data availability

The DIV2K dataset was used for this study [https://data.vision.ee.ethz.ch/cvl/DIV2K/][33]. Raw data underlying this work can be accessed through the figshare repository, available at [https://figshare.com/projects/MAI-SIM/140008]. Source data are provided with this paper.

## Code availability

All the code used for image processing and hardware control is available from the GitHub repository for the project [https://github.com/LAG-MNG-CambridgeUniversity/MAI-SIM][32].

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

## Acknowledgements

The authors would like to acknowledge Florian Ströhl and James Manton for their insights and contributions to the design. The authors would also like to thank Anna Scheeder, Katharina Scherer, Nino Läubli, and Rebecca McClelland for their helpful comments on the manuscript. This work was performed using resources provided by the Cambridge Service for Data Driven Discovery (CSD3) operated by the University of Cambridge Research Computing Service, provided by Dell EMC and Intel using Tier-2 funding from the Engineering and Physical Sciences Research Council (capital grant EP/T022159/1), and DiRAC funding from the Science and Technology Facilities Council.

## Author contributions

E.N.W. and L.H. designed and constructed the system, developed the image processing software, and collected data. C.N.C., E.N.W., and J.R.L. developed the machine-learning reconstruction. A.W. and J.R.L. contributed to the design of the setup. A.W. and E.N.W. developed the iterative reconstruction method. C.W.C., L.M., and M.L. prepared fixed and live biological samples. C.F.K. and C.J.R. conceived the interferometry method. C.F.K. and G.S.K. supervised the project. E.N.W., L.H., and C.F.K. wrote the manuscript. C.F.K. acknowledges funding from the UK Engineering and Physical Sciences Research Council (EP/L015889/1 and EP/H018301/1), the Wellcome Trust, (3-3249/Z/16/Z and 089703/Z/09/Z) the UK Medical Research Council (MR/K015850/1 and MR/K02292X/1), and Infinitus Ltd.

## Competing interests

The authors declare no competing interests.
