## [Peer Review File · Nature Communications]

REVIEWER COMMENTS

Reviewer #1 (Remarks to the Author):

The paper presents a SIM implementation based on Michelson interferometer. This configuration enables simultaneous multi-color SIM live cell imaging, in contrast, current SLM- or DMD-based solutions have to produce SIM super-res images sequentially. Moreover, the authors apply their previously published ML-SIM for SIM image reconstruction. Overall, this work proposes a novel implementation for SIM, I support the publication for this paper, but given that the following points has been addressed.

(1) The authors claim MAI-SIM achieves real-time implantation, but the reconstruction rate is around 1Hz (the 3rd paragraph in Discussion). It is well-established that SLM and sCMOS camera-based SIM can reach tens to hundreds Hz imaging speed. I wonder what is the limit speed of interferometer based SIM implementation.

(2) The resolution is identified FWHM. However, iterative deconvolution usually generates the structures smaller than the theoretical limit of SIM. It is preferred to measure the resolution using multiple state-of-the-art methods, such as FRC and decorrelation.

(3) The author has noticed that the phase step in MAI-SIM is uneven because of structured illumination spacing mismatching. However, how the uneven phase step affects the reconstruction have not been characterized. Because the condition number of the corresponding separation matrix may be larger than the even phase step, the separation and afterwards reconstruction might be prone to artifacts.

(4) Two comments: a) the author directly applied previously developed ML-SIM deep learning reconstruction algorithm into MAI-SIM, I cannot identify any innovation in this part. b) the authors state that "For the case of uneven phase steps, traditional Fourier based reconstruction techniques do not perform optimally". In fact, the traditional Fourier-based reconstruction technique can deal with uneven phase steps if the corresponding phase steps are assigned to construct the separation matrix.

(5) The deep learning part is kind of hard to follow. First, the schematic of training/test should be provided. Second, the natural image from DIV2K dataset is used to train the model, can you prove the synthetic degradation is the same as real SIM experiments. Although the domain gap can be reduced by transfer learning, it still needs high-quality 100-nm-resolution fluorescence images as training targets. Since the authors has developed the new system, I suggest the authors do a systematic evaluation using experimental data.

Reviewer #2 (Remarks to the Author):

In their manuscript entitled "MAI-SIM: interferometric multicolor structured illumination microscopy for everybody" the authors Edward N. Ward et al. present both an Michelson-interferometer based approach to implement the illumination arm of a SIM microscope, as well as a machine-learning based data processing pipeline optimized for the post-processing needs of this system.

The implementation by means of a Michelson interferometer is quite elegant, as it allows to create the classic Gustafsson-style sinusoidal SIM illumination with only a galvanometric mirror as a single moving element. The authors demonstrate that in color-sequential mode, this approach generates high-quality SIM data that can be handled with 'classical' established SIM reconstruction algorithms. To me, this finding is important (and maybe can be stressed more in the manuscript) for those users skeptical of machine-learning based data processing.

The post-processing through a machine-learning based data pipeline is mostly presented as a means to deal with the uneven SIM phase shifts inherent to the color-parallel mode, though in my experience also aspects of denoising and the avoidance of classical SIM artifacts are handled well by these machine-learning models. The authors seem to take the necessary care (lines 158 to 160) to avoid these networks from introducing artifacts on their own.

Both the instrument design and post-processing pipeline are made available as 'open science'

results in online repositories, which I highly appreciate.

I support publication of the manuscript.

I have a few questions and suggestion for potential improvements:

- * The description of typical SIM implementations (lines 45ff) is correct, but lacking the implementation of the popular GE OMXv4 and OMX-SR instruments, which use a combination of galvanometric mirrors (and glass plates) to steer and phase-shift the interfering beams. While more complex than the solution presented here, the systems share many of the advantages, especially concerning imaging speed. Indeed, the rotation system (by switching to differently oriented mirror pairs) is quite similar in the OMX and in the MAI-SIM system.
- * For SLM-based SIMs, indeed non-integer sized phase shifts are possible (by rotating the pattern but shifting horizontally/vertically, see R. Heintzmans search algorithms), and passive segmented polarization control is easily possible for 2-beam SIM. I would stress light loss and maybe a somewhat limited field of view as the main drawbacks of SLM-based systems.
- * For the resolution estimation in Fig. 2, I'd recommend to provide a Fourier Ring Correlation or Image Decorrelation Analysis (A Descloux, Nat. Meth 16, pages 918–924 (2019)) based estimation and plot, as cross-section plots are very susceptible to filtering artifacts in SIM.
- * Reconstruction with non-equidistant phases, and indeed with raw data based absolute phase estimation for every reconstructed frame, is indeed possible with fairSIM, though this feature is only accessible when using a current version built from source code. This feature might offer enhanced reconstruction quality in the color-parallel mode, though I do not expect it to outperform the machine learning based reconstruction.
- * While I fully expect GPU-based live reconstruction to be possible with a machine-learning based reconstruction approach such as MAI-SIM (after all, machine learning frameworks are highly optimized and the networks are not too complex), GPU-acceleration also makes classic SIM reconstruction possible in real time (A. Markwirth et. al, Nat. Comm 10, 4315 (2019), with 2x 30fps 512x512 on a GTX 690), and I'd rather state the GPU-acceleration (of a problem suitable for massively parallel computing) is what realized the speed gain.

Response to reviewer comments

Reviewer #1:

The paper presents a SIM implementation based on Michelson interferometer. This configuration enables simultaneous multi-color SIM live cell imaging, in contrast, current SLM- or DMD-based solutions have to produce SIM super-res images sequentially. Moreover, the authors apply their previously published ML-SIM for SIM image reconstruction. Overall, this work proposes a novel implementation for SIM, I support the publication for this paper, but given that the following points has been addressed.

(1) The authors claim MAI-SIM achieves real-time implantation, but the reconstruction rate is around 1Hz (the 3rd paragraph in Discussion). It is well-established that SLM and sCMOS camera-based SIM can reach tens to hundreds Hz imaging speed. I wonder what is the limit speed of interferometer based SIM implementation.

We have been unclear about this in our writing. The 1Hz rate we achieve only relates to the machine learning reconstruction process on the hardware we had available for the work presented. Indeed, the raw imaging speed is higher also on our system and we report frame rates of ~3Hz (Fig. 7) for 3 colours simultaneously, a limit imposed by our camera speed at full field of view. Frame rates up to 4.5 Hz are possible with smaller fields of view, for example in the case of single colour imaging. In this instance, the physical limit for imaging speed in our system is imposed by the settling time of the galvo mirror. The mirror element on our galvo is unnecessarily large and requires settling times of ca. 20 ms. We are aware of commercially available galvo mirrors requiring only several μ s to settle and this would afford significant gains in speed (i.e. only limited by camera frame rates at all FOVs), indeed reaching the speeds referred to by the reviewer. However, we note that are able to image in multiple colours simultaneously at optimal resolution, which already offsets the reduced frame rates and is an advantage not afforded by other methods. We have clarified these points in the discussion, highlighted the limitation of the current system, and have now described a method for increasing frame rates in the future.

We have included the following text in the Discussion section on page 13:

The maximum imaging speed of our system is limited by the settling time galvanometric mirror element used. Because of its size and associated inertia it requires several tens of milliseconds to settle between mirror-to-mirror jumps. The system could be readily adapted to make use of a smaller, and faster, mirror element so long as a de-magnified image of the objective back focal plane does not exceed the size of the mirror. With recently available MEMS mirrors, settling should not be the limiting parameter of imaging speed and frame rates only limited by camera speed would then be available for MAI-SIM.

(2) The resolution is identified FWHM. However, iterative deconvolution usually generates the structures smaller than the theoretical limit of SIM. It is preferred to measure the resolution using multiple state-of-the-art methods, such as FRC and decorrelation.

We agree with this comment. As stated in the manuscript, the reported resolution increases were indeed compared to deconvolved wide field images (figure 2). For a quantitative measure of resolution we have now performed a Fourier Ring Correlation (FRC) analysis which confirms our pated resolution increases. These are now included as a new supplementary figure (Suppl. Fig. S7).

(3) The author has noticed that the phase step in MAI-SIM is uneven because of structured illumination spacing mismatching. However, how the uneven phase step affects the reconstruction have not been characterized. Because the condition number of the corresponding separation matrix may be larger than the even phase step, the separation and afterwards reconstruction might be prone to artifacts.

It is known that uneven phase steps lead to reconstruction artefacts in classical reconstruction methods, and this is the reason we have developed a machine learning framework to deal precisely with this issue. A full discussion of this problem, including a test of *phase error on reconstruction fidelity* is given in section 6 of the supplementary section in our previous article on ML reconstruction methods in SIM (main text, ref. [14]). There is also a specific section in the current manuscript (table S1, Supplementary Material) where we measure the effect of phase errors on reconstruction. To clarify this further, we have now produced simulated SIM data obtained with various levels of phase errors and quantified the resulting errors using the different reconstruction methods (SI Fig. S9).

We refer to this new section in the main text as follows (page 9):

We include a comparison of reconstruction fidelity of for various SIM algorithms as a function of phase errors during phase stepping (SI Figure S9). Even for very large phase errors, ML SIM produces reconstructions without artefacts, which is in contrast to other methods, which become susceptible to pattern artefacts in reconstructed images. For the imaging of dynamically changing and light sensitive samples, where short camera exposure times and low excitation intensities have to be used, the ability to handle noise and uneven phase stepping presents significant advantages.

(4) Two comments: a) the author directly applied previously developed ML-SIM deep learning reconstruction algorithm into MAI-SIM, I cannot identify any innovation in this part. b) the authors state that “For the case of uneven phase steps, traditional Fourier based reconstruction techniques do not perform optimally”. In fact, the traditional Fourier-based reconstruction technique can deal with uneven phase steps if the corresponding phase steps are assigned to construct the separation matrix.

a. Machine Learning reconstruction: As stated in the manuscript, the ML reconstruction component is based on the approach reported earlier (main text, ref. [14] and [19]). We have retrained the network specifically to deal with uneven phase steps for the current work to increase robustness in reconstruction fidelity. The biggest novelty is however the implementation of the high-performance real-time reconstruction capability that enables live viewing of reconstructed MAI-SIM data in multiple colours during recording. To enable real-time reconstruction, a completely new pipeline had to be developed, integrating hardware control and reconstruction whilst the system records live data. We note that the generality of the framework makes it applicable to many other machine learning tasks, such as de-noising and segmentation. All code is available on Github and we hope it will be useful in many other contexts to other research groups. We have now clarified this in the supplementary text and added a figure to describe the network training algorithm (supplementary figure S8 and corresponding text).

b. It is true that Fourier methods are, theoretically, able to reconstruct images with uneven phase steps if the separation matrix is assembled correctly. Indeed, this is the basis of the inverse matrix approach. However, as the phase steps become more uneven, the condition number of the separation matrix increases as it approaches a singular matrix. In practice, this makes the reconstructions much more prone to artefacts from noise in the data. As ML-SIM is trained specifically on data with uneven phase steps, it maintains high quality reconstructions even with highly uneven phase steps. We have now included a new figure (SI Fig. S9) to demonstrate this.

(5) The deep learning part is kind of hard to follow. First, the schematic of training/test should be provided. Second, the natural image from DIV2K dataset is used to train the model, can you prove the synthetic degradation is the same as real SIM experiments. Although the domain gap can be reduced by transfer learning, it still needs high-quality 100-nm-resolution fluorescence images as training targets. Since the authors has developed the new system, I suggest the authors do a systematic evaluation using experimental data.

Regarding the first point, we have now included a new figure to show the pipeline for data generation and network training (SI Fig. S8). We disagree with the second point on using real SIM data as ground truth. The problem of generating useful *experimental* training data for SIM reconstruction is fraught with problems and precisely the reason for the rising popularity of our transfer learning approach in the SIM community (main text, ref [14]). There exists no way to generate a suitably diverse training dataset with ground truth targets that are free from imaging artefacts. The optimal approach combines physical models, as used in classical reconstruction algorithms, with errors that are representative of real experimental setups to produce training data from artefact free ground truth images. The relative merits of transfer learning generally, and the point raised by the reviewer specifically, have been discussed in detail in our previous publications (main text, ref. [14] and [19]).

Reviewer #2:

In their manuscript entitled “MAI-SIM: interferometric multicolor structured illumination microscopy for everybody” the authors Edward N. Ward et al. present both an Michelson-interferometer based approach to implement the illumination arm of a SIM microscope, as well as a machine-learning based data processing pipeline optimized for the post-processing needs of this system.

The implementation by means of a Michelson interferometer is quite elegant, as it allows to create the classic Gustafsson-style sinusoidal SIM illumination with only a galvanometric mirror as a single moving element. The authors demonstrate that in color-sequential mode, this approach generates high-quality SIM data that can be handled with ‘classical’ established SIM reconstruction algorithms. To me, this finding is important (and maybe can be stressed more in the manuscript) for those users skeptical of machine-learning based data processing.

Thank you for this useful comment. We have now highlighted this point more explicitly in the discussion section and added the following in the Discussion section (page 13):

MAI-SIM is straightforward and cost-effective to implement on existing widefield microscope frames. Furthermore, our method does not fundamentally differ from conventional SLM or DMD based SIM systems in the way it shifts high frequency sample information into the optical pass band. As a result, all classical reconstruction methods are in principle applicable to our method if phase stepping is sufficiently well controlled during imaging.

The post-processing through a machine-learning based data pipeline is mostly presented as a means to deal with the uneven SIM phase shifts inherent to the color-parallel mode, though in my experience also aspects of denoising and the avoidance of classical SIM artifacts are handled well by these machine-learning models. The authors seem to take the necessary care (lines 158 to 160) to avoid these networks from introducing artifacts on their own.

Both the instrument design and post-processing pipeline are made available as ‘open science’ results in online repositories, which I highly appreciate.

I support publication of the manuscript.

I have a few questions and suggestion for potential improvements:

**** The description of typical SIM implementations (lines 45ff) is correct, but lacking the implementation of the popular GE OMXv4 and OMX-SR instruments, which use a combination of galvanometric mirrors (and glass plates) to steer and phase-shift the interfering beams. While more complex than the solution presented here, the systems share many of the advantages, especially concerning imaging speed. Indeed, the rotation system (by switching to differently oriented mirror pairs) is quite similar in the OMX and in the MAI-SIM system.***

We have now included these methods with a brief comparison to our own method in the hardware description (page 4).

Our implementation greatly reduces the complexity associated with alternative interferometric SIM systems, such as that employed in the latest iteration of the commercially available OMX SIM microscope. Here one

requires multiple moving optical elements and piezo stacks for phase stepping.¹⁰⁻¹² The required coherence length for lasers in our system is 10s of micrometers, rather than 10s of cm in alternatives and keeping phase stability is facilitated by the shorter relative path differences in MAI-SIM.

**** For SLM-based SIMs, indeed non-integer sized phase shifts are possible (by rotating the pattern but shifting horizontally/vertically, see R. Heintzmans search algorithms), and passive segmented polarization control is easily possible for 2-beam SIM. I would stress light loss and maybe a somewhat limited field of view as the main drawbacks of SLM-based systems.***

We have now included the following paragraph in the Introduction (page 2):

Additionally, the discrete number and finite size of pixels restrict the possible periodicities of the pattern and introduce unwanted diffraction orders, which reduces photon efficiencies.³ Typically light throughput in such systems is limited to a few percent of the input energy, requiring high power lasers for illumination. The Michelson implementation has a further advantage of permitting modulation patterns to be produced over large fields of view, since they are not restricted by the physical size / number of pixels of the SLM/DMD.

**** For the resolution estimation in Fig. 2, I'd recommend to provide a Fourier Ring Correlation or Image Decorrelation Analysis (A Descloux, Nat. Meth 16, pages 918–924 (2019)) based estimation and plot, as cross-section plots are very susceptible to filtering artifacts in SIM.***

Thank you for this point, also made identically by reviewer 1. We agree and have included these extra data - see reply to comment 2 of reviewer 1.

**** Reconstruction with non-equidistant phases, and indeed with raw data based absolute phase estimation for every reconstructed frame, is indeed possible with fairSIM, though this feature is only accessible when using a current version built from source code. This feature might offer enhanced reconstruction quality in the color-parallel mode, though I do not expect it to outperform the machine learning based reconstruction.***

We have begun to try this with FairSIM, but encountered significant difficulties compiling the publicly available source code with this functionality as it is unstable and under continuing development. We will work with the authors of the code from the Marcel Mueller lab to attempt this in future.

**** While I fully expect GPU-based live reconstruction to be possible with a machine-learning based reconstruction approach such as MAI-SIM (after all, machine learning frameworks are highly optimized and the networks are not too complex), GPU-acceleration also makes classic SIM reconstruction possible in real time (A. Markwirth et al, Nat. Comm 10, 4315 (2019), with 2x 30fps 512x512 on a GTX 690), and I'd rather state the GPU-acceleration (of a problem suitable for massively parallel computing) is what realized the speed gain.***

We agree with this point in principle, but there are conceptual differences that are advantages for ML SIM in terms of speed. In classic reconstruction, the time limiting step is set by the requirement to estimate frequency and phase of the input modulation pattern. The reported speeds achievable in other “real-time” reconstruction implementations are only achievable if a pattern has already been estimated accurately under the assumption that it remains invariant during imaging. To our

knowledge, reconstruction *and* pattern estimation cannot be performed simultaneously at video rate speeds, even with GPU acceleration. For experiments where the modulation patterns change (even subtly), this will produce artefacts. Because this limitation does not exist in ML SIM, reconstruction quality is maintained at very high speed, even when modulation patterns start to change during imaging, e.g. due to sample movement, refractive index changes, mechanical shifts, etc.

Appendix

New figures added to text

Figure S7: Resolution estimation with Fourier ring correlation: A: Widefield and SIM images of 100 nm fluorescent Tetraspeck spheres imaged at 647 nm and 561 nm excitation wavelengths (emission captured around 680 nm and 580 nm, respectively). SIM images were reconstructed using the inverse matrix approach. Scale bar corresponds to 1 μm . B: Fourier Ring Correlation (FRC) curves for the reconstructed microsphere images. The resolution is estimated from the intersection of the correlation curves (red and green) with the 3-sigma curve (orange). The ca. 1.77 times resolution increase is in agreement with the resolution increase predicted from the spacing between fringes in the illumination pattern. FRC measurements were performed using the FRC plugin in ImageJ.

Figure S8: Schematic of data generation and network training for ML-SIM. Training data is generated from the DIV2K image dataset. Single images are taken randomly from the dataset undergo a greyscale conversion followed by resizing, rotation and thresholding to achieve a modal pixel value of 0. This has the effect of introducing high resolution features into the images as well as making them visually more similar to typical fluorescence microscopy images. Simulated SIM images are generated from the ground truth by multiplication with nine sinusoidal SIM fringes and convolution with an in-focus PSF. Crucially, and unlike previous ML-SIM implementations, these SIM fringes are generated with a high degree of phase shift error to simulate data acquired on the MAI-SIM system in simultaneous acquisition mode. A mixed Poisson-Gaussian noise model is then used to simulated photon shot noise and camera readout noise. Finally, salt and pepper noise is added to simulate hot pixels and dead pixels respectively. Details on the parameters used for generating the dataset can be found in the documentation in the project repository. This data generation pipeline is used to generate a training dataset consisting of 6000 simulated SIM images and the ML-SIM model is then trained by iterating through this dataset for 200 epochs. The advantages of this transfer learning approach are that very large datasets can be generated to ensure generalization and ideal ground truth images can be used to prevent the model from learning to replicate the imaging artefacts inevitably present in experimental datasets.

Figure S9: Reconstruction performance of various SIM algorithms in the presence of phase errors during patterns changes. A: Left panel: Ground truth image used to simulate the effect of phase errors on SIM reconstruction fidelity. The SIM raw data were calculated from this image in the presence of mixed Poisson and Gaussian noise (ca 15 percent rms noise). Raw images, were calculated for phases 0 , $120^\circ - \Delta\phi$, and $240^\circ - 2\Delta\phi$, where $\Delta\phi$ is the phase error. Right panels: Comparison of images reconstructed with various methods on simulated SIM raw data obtained using a phase error of $\Delta\phi = 107^\circ$. B: Structural similarity (SSIM) scores for the reconstructions measured against the ground truth image for increasing phase errors $\Delta\phi$. SSIM measurements were performed in ImageJ using the available plugin.

REVIEWERS' COMMENTS

Reviewer #1 (Remarks to the Author):

The improvements and response in the latest manuscript have addressed my concerns, looking forward to seeing this work in the open-source community.

Reviewer #2 (Remarks to the Author):

The authors provided a thorough reply to the questions raised in the initial review, and revised the manuscript. I consider the points I have raised in the initial review well answered and addressed, and recommend publication of the manuscript.